# Gut Microbial Diversity Reveals Differences in Pathogenicity between *Metarhizium rileyi* and *Beauveria bassiana* during the Early Stage of Infection in *Spodoptera litura* Larvae

**DOI:** 10.3390/microorganisms12061129

**Published:** 2024-05-31

**Authors:** Guang Wang, Sicai Xu, Laiyan Chen, Tianjiao Zhan, Xu Zhang, Honghui Liang, Bin Chen, Yuejin Peng

**Affiliations:** Yunnan State Key Laboratory of Conservation and Utilization of Biological Resources, College of Plant Protection, Yunnan Agricultural University, Kunming 650201, China; wang_guang_17@126.com (G.W.); 15198649252@163.com (S.X.); 18788118783@163.com (L.C.); pochama0393@163.com (T.Z.); zhxv08@163.com (X.Z.); m18082928395@163.com (H.L.)

**Keywords:** pathogenicity difference, *Beauveria bassiana*, *Metarhizium rileyi*, gut microbes, host defense

## Abstract

*Beauveria bassiana* and *Metarhizium rileyi* are extensively utilized to investigate fungal pathogenic mechanisms and to develop biological control agents. Notwithstanding, notable distinctions exist in their pathogenicity against the same host insect. This study aimed to elucidate the pathogenic differences between *M. rileyi* and *B. bassiana* by examining the impact of various ratios of *B. bassiana* strain AJS91881 and *M. rileyi* strain SXBN200920 on fifth instar larvae of *Spodoptera litura*, focusing on early infection stages and intestinal microbial community structure. The lethal time 50 (LT_50_) for *B. bassiana* was significantly lower than that for *M. rileyi*, indicating greater efficacy. Survival analyses in mixed groups (ratios of 1:9, 1:1, and 9:1 *M. rileyi* to *B. bassiana*) consistently demonstrated higher virulence of *B. bassiana*. Intestinal microbial diversity analysis revealed a significant increase in *Achromobacter* and *Pseudomonas* in larvae infected with *M. rileyi*, whereas *Weissella* was notably higher in those infected with *B. bassiana.* Additionally, significant shifts in microbial genera abundances were observed across all mixed infection groups. KEGG pathway enrichment analysis indicated that *M. rileyi* and *B. bassiana* employ distinct pathogenic strategies during early infection stages. *In vitro* tests confirmed the superior growth and stress resistance of *B. bassiana* compared to *M. rileyi,* but the antifungal ability of *M. rileyi* was better than that of *B. bassiana.* In conclusion, our findings provide preliminary insights into the differential pathogenic behaviors of *M. rileyi* and *B. bassiana* during the early infection stages in *S. litura* larvae, enhancing our understanding of their mechanisms and informing biological pest control strategies in agriculture and forestry.

## 1. Introduction

Entomopathogenic fungi, such as *Beauveria bassiana* [1], *Metarhizium rileyi* [2], and *Metarhizium anisopliae* [3], are microorganisms capable of specifically infecting and killing insects. Research has identified numerous factors influencing fungal pathogenicity, including growth and development [4], evasion of host immunity [5], and resistance to environmental stresses such as UV radiation, high temperatures, and oxidative stress [6].

The infection process of entomopathogenic fungi varies due to the different physiological structures of their hosts. The infection cycle of *Metarhizium* spp. encompasses four stages: adhesion to the insect's body wall, germination to form a bud tube, development of appressoria and penetration of the insect's body wall, and colonization of the insect’s hemocoel followed by saprophytic growth on insect cadavers [7]. The formation of appressoria by *Metarhizium* spp. is crucial for breaching the insect body wall. During this stage, fungal spores amass various nutrients such as lipids, glycogen, polyols, and trehalose [8]. These lipids are transported to the appressorium where they are metabolized into glycerol, which increases turgor pressure within the appressorium, thereby enhancing the mechanical force required to penetrate the insect body wall [9]. Conversely, when *Beauveria* spp. infect a host, they breach the insect body surface directly through a germination tube that exerts mechanical pressure and secrete a variety of enzymes that degrade the body surface [10].

After infection, the mycelia of pathogenic fungi undergo a dimorphic transformation from filamentous fungal cells to yeast-like spores, proliferating within the host [7]. Studies indicate that fungi of the *Beauveria* and *Metarhizium* genera adopt distinct pathogenic strategies. For instance, *Metarhizium* spp. not only proliferate rapidly by absorbing nutrients from the host but also secrete destruxin to facilitate host mortality [11]. *Beauveria* spp. produce beauvericin [12], oosporein [13], and other insecticidal compounds, enhancing their proliferation and stress resistance capabilities.

The invasion of entomopathogenic fungi and their interaction with pathogenic hosts are complex processes. Typically, the host microbiome maintains a dynamic ecological balance. However, once invaded by entomopathogenic fungi, this microecological balance is disrupted, leading to complex interactions among the host, the pathogenic fungus, and host microbial communities [14]. Insects provide diverse habitats for microorganisms, including their surface, intestines, hemocoel, and cells [15]. The gut microbiota plays a crucial role in digestion, immune response, and disease transmission [16], and the dynamic balance of this microbial community is an important indicator of host health.

The first sentence of Leo Tolstoy's (2001) novel Anna Karenina is: “Happy families are all alike; every unhappy family is unhappy in its own way”, calling it the Anna Karenina principle (AKP). Applying the APK to animal microbiomes, stressors reduce the ability of the host or its microbiome to regulate community composition [17]. Recent research has demonstrated that pathogenic fungi interact with the gut flora, influencing insect health and longevity [18,19]. During fungal infections, while changes occur in the gut microbiome, the microbiota employs various strategies to counter the infection and resist defenses. For example, it can eradicate pathogenic fungi by activating the host’s fluid antimicrobial immunity, which prevents fungi from competing for nutrients in the hemolymph [20]. Additionally, under adverse environmental conditions or when the host is in a pathological state, gut microbes that are typically symbiotic may become opportunistic pathogens [21,22]. 

However, the intestinal microbial community structure of the same insect species infected by different pathogenic fungi varies significantly [18,23]. These findings suggest that various pathogenic fungi may employ distinct strategies when competing with host intestinal microbes. Yet, comparative analyses of the pathogenic process in this context are sparse. In this study, different proportions of *M. rileyi* and *B. bassiana* were used to infect third instar larvae of *S. litura*, and changes in their intestinal microbial communities were compared to elucidate the mechanisms underlying the differences in virulence between *M. rileyi* and *B. bassiana* during hemocoelic infection. The results will enhance understanding of the pathogenic mechanisms of various fungi and inform the biological control of agricultural pests.

## 2. Materials and Methods

### 2.1. Fungi and Media

*M. rileyi* strain XSBN200920 was collected from infected *S. frugiperda* in Xishuangbanna Dai Autonomous Prefecture (22°10′22″ N, 100°51′29″ E), Yunnan Province, China, in 2019, as previously described [18]. *B. bassiana* strain AJS91881 was isolated from infected *S. litura* in Lancang County (22°37′20″ N, 99°57′57″ E), Yunnan Province, China, in 2019 [6], as previously described. Both fungal strains were maintained in our laboratory. *M. rileyi* and *B. bassiana* were cultured in Sabouraud maltose agar medium (SMAY: 1% peptone, 1% yeast extract, 4% maltose, and 1.5% agarose) plates [24] and Sabouraud dextrose agar medium (SDAY: 1% peptone, 1% yeast extract, 4% glucose, and 1.5% agar) plates [25], respectively. These fungi were incubated at 25°C under a 12-hour light/12-hour dark cycle in a culture chamber. Conidial suspensions were prepared from fungi grown on SMAY and SDAY plates for 10 days.

### 2.2. Insect Feeding

The laboratory population of *S. litura* was reared indoors in artificial climate chambers set at 25°C, with a photoperiod of 16L:8D and relative humidity (RH) of 70%. Larvae were fed a fresh artificial diet (125 g soybean flour, 225 g corn flour, 225 g corn leaf flour, 40 g yeast extract, 20 g casein, 0.6 g cholesterol, 3 g choline chloride, 6 g sorbic acid, 6 g methyl p-hydroxybenzoate, 36 g agarose, 0.1 g inositol, 7 g vitamin C, and 1300 mL water) daily. For adult rearing, adults were kept in a cage (50 × 50 × 50 cm), and fresh honey water was provided every other day.

### 2.3. Bioassay

For the intrahemocoelic injection assay, fifth instar larvae of *S. litura*, cultured for over five generations, were used to determine virulence. The larvae were randomly divided into five groups: (1) 1:0, *S. litura* receiving 10^5^ conidia/mL suspension with *M. rileyi*; (2) 9:1, *S. litura* receiving 10^5^ conidia/mL suspension with 9:1 *M. rileyi* : *B. bassiana* (V/V); (3) 1:1, *S. litura* receiving 10^5^ conidia/mL suspension with 1:1 *M. rileyi* : *B. bassiana* (V/V); (4) 1:9, *S. litura* receiving 10^5^ conidia/mL suspension with 1:9 *M. rileyi* : *B. bassiana* (V/V); and (5) 0:1, *S. litura* receiving 10^5^ conidia/mL suspension with *B. bassiana.* In total, 5 μL of the conidia suspension was injected into the haemocoel of each larva in each group. Tween-80 water (0.02%) used for the suspension of blastospores was injected as a control. This experiment was conducted three times with no fewer than 35 samples per treatment. Post-treatment, larvae were maintained by feeding them an artificial diet in an artificial climate chamber at 25 °C, 75% relative humidity, and a 16:8 light/dark cycle. Mortality was monitored every 12 h, and all deceased larvae were placed in a Petri dish to monitor moisture levels and confirm infection by the test strains. The dead insects were placed in a culture dish for about 10 days and photographed.

To examine in vivo hyphal body growth, each fifth instar larvae was injected with the 1:0 and 0:1 groups of the conidial suspension (5 μL, 1 × 10^5^ conidia/mL), and the infected larvae were fed an artificial diet in an artificial climate chamber at 25 °C, 75% relative humidity, and a 16:8 light/dark cycle. The hemolymph of each group with 5 larvae was collected, which was diluted 1:1 in sterile anticoagulant (0.14 M NaCl, 0.1 M glucose, 25 mM sodium citrate, and 30 mM citric acid). Hyphal bodies were photographed and observed at 60 and 132 h post-injection (HPI) of the conidia under a microscope.

### 2.4. Physiology Experiments

Aliquots of 2.5 μL conidial suspension (1 × 10^7^ spore/mL) from *M. rileyi* strain XSBN200920 (1:0 group) and *B. bassiana* strain AJS91881 (0:1 group) were inoculated into Czapek–Dox Medium (CZA: 0.1% dipotassium phosphate, 0.05% potassium chloride, 0.0001% ferrous sulfate, 0.3% sodium nitrate, 0.05% magnesium sulfate, 3% sucrose, and 1.5% agar) containing 0.5 M/L NaCl, 50 μg/mL Congo Red, 0.03 mM/L Menadione, and 3 mM/L H_2_O_2_. Plates were incubated at 25 °C under a 12-h light/dark cycle within a culture chamber at 25 °C under a 12-h light/12-h dark cycle. Colonies were photographed after 7 days of incubation. In the *M. rileyi* and *B. bassiana* plate confrontation experiment, the SMAY plate was used as a vegetative growth medium for two strains of fungi. In total, 2.5 µL of 1 × 10^7^ spores/mL fungal conidia suspension was inoculated on the plate and photographed on day 10 of growth. The spore distance of the two strains was 4 cm and 2.5 cm, respectively. The experiment was repeated in triplicate, with three parallel controls each time.

### 2.5. DNA Extraction and PCR Amplification

In the treatment group, a 1.0 × 10^5^ conidia/mL suspension was injected into the fifth instar larvae of *S. litura*. Five days post-injection, the intestines of the injected larvae were carefully dissected. Genomic DNA of the insect gut microorganisms was isolated with assistance from Majorbio Co., Ltd., Shanghai, China. Total microbial genomic DNA was extracted using the E.Z.N.A.^®^ Soil DNA Kit (Omega Bio-tek, Norcross, GA, USA) following the manufacturer’s instructions. The quality and concentration of DNA were assessed using 1.0% agarose gel electrophoresis and a NanoDrop^®^ ND-2000 spectrophotometer (Thermo Scientific Inc., Wilmington, DE, USA), and samples were stored at −80 °C for subsequent analyses. The 16S rRNA gene was amplified using an ABI GeneAmp^®^ 9700 PCR thermocycler (ABI, Foster, CA, USA). The PCR mixture included 4 µL of 5× Fast Pfu buffer, 2 µL of 2.5 mM dNTPs, 0.8 µL of each primer (5 µM), 0.4 µL of Fast Pfu polymerase, 10 ng of template DNA, and ddH_2_O to a final volume of 20 µL. PCR conditions were as follows: an initial denaturation at 95 °C for 3 min, followed by 27 cycles of denaturation at 95 °C for 30 s, annealing at 55 °C for 30 s, and extension at 72 °C for 45 s, with a final extension at 72 °C for 10 min, and ending at 4 °C. Three replicates were processed for each sample. The PCR products were extracted from a 2% agarose gel, purified using the AxyPrep DNA Gel Extraction Kit (Axygen Biosciences, Union, CA, USA) following the manufacturer’s instructions, and quantified with a Quantus™ Fluorometer (Promega, Madison, WI, USA).

### 2.6. Illumina MiSeq Sequencing

Purified amplicons were pooled in equimolar ratios and sequenced using paired-end reads on an Illumina MiSeq PE300 platform (Illumina, San Diego, CA, USA) following standard protocols by Majorbio Bio-Pharm Technology Co., Ltd. (Shanghai, China).

### 2.7. Data Processing

Raw FASTQ files were de-multiplexed using an in-house Perl script and then quality-filtered with fastp version 0.19.6 and merged using FLASH version 1.2.7. The optimized sequences were clustered into operational taxonomic units (OTUs) using UPARSE 7.1 at a 97% sequence similarity level. For each OTU, the most abundant sequence was selected as the representative sequence. All singletons were removed from the OTU dataset. Chloroplast sequences were manually removed from all samples in the OTU table. To minimize the effects of sequencing depth on alpha and beta diversity measures, the number of 16S rRNA gene sequences from each sample was rarefied to 20,000, achieving an average Good’s coverage of 99.09%. OTUs were aligned using the Silva database to obtain taxonomic information. Again, the most abundant sequence for each OTU was chosen as the representative sequence.

Bioinformatic analysis of the gut microbiota was conducted on the Majorbio Cloud platform (https://cloud.majorbio.com, accessed on 23 March 2024). Based on the OTUs, rarefaction curves and alpha diversity indices, including observed OTUs, were generated. A Kruskal–Wallis rank-sum test was used to compare the relative abundance of intestinal microbes at the phylum and genus levels between groups. The α multiplicity-Chao richness estimator was calculated using QIIME2 (https://qiime2.org/, accessed on 23 March 2024), reflecting the species richness and community diversity. The similarity among microbial communities in different samples was assessed by principal component analysis (PCA) based on Bray-Curtis dissimilarity using the Vegan v2.5-3 package. Species with significant differences in sample classification were identified using linear discriminant analysis (LDA ≥ 2, *p* < 0.05). Tax4Fun, which predicts functional profiles from metagenomic 16S rRNA data, was employed to derive three levels of metabolic pathway information and pathway abundance [26]. Additionally, ternary plots, the average variation degree (AVD), and a correlation network were analyzed on the Majorbio Cloud platform.

### 2.8. Phenotypes of Fungal Stress and Growth

All data were represented as the means ± SD of three replicates per treatment. Percentage of relative growth inhibition (RGI) was estimated with the formula (*d*_c_ − *d*_t_)/*d*_c_ × 100 (*d*_c_, control colony diameter; *d*_t_, stressed colony diameter) and used as an index of hyphal sensitivity to each stress. The survival data were subjected to a Kaplan–Meier survival log-rank analysis. The median lethal time (LT_50_) for certain treatments was calculated using linear regression in Microsoft Excel version 2019. Data conforming to a normal distribution and homogeneity of variance between two groups were compared using an unpaired t-test. Comparisons among multiple groups were conducted via a univariate analysis of variance (ANOVA) and bivariate ANOVA, followed by Tukey’s honest significance difference test (Tukey’s HSD). When using parametric tests, a Gaussian distribution of data was checked using Shapiro–Wilk normality tests. A *p*-value of <0.05 was considered statistically significant. 

## 3. Results

### 3.1. Differences in Virulence in Mixtures of M. rileyi and B. bassiana

Virulence test results indicated that the survival rates of fifth instar larvae treated with *M. rileyi* and *B. bassiana* in 1:1, 1:9, and 0:1 ratios were significantly lower (log-rank test, *p* < 0.0001) than those treated at the 1:0 and 9:1 ratios (Figure 1A). Larval mortality commenced at 3.5 days in the 1:1, 1:9, and 0:1 groups, at 4.5 days in the 9:1 group, and at 6 days in the 1:0 group (Figure 1A). The median lethal times for *M. rileyi* and *B. bassiana* in the 1:0, 9:1, 1:1, 1:9, and 0:1 groups were 9.21 ± 0.39, 8.38 ± 0.15, 4.84 ± 0.18, 4.90 ± 0.10, and 4.66 ± 0.50 days, respectively (Figure 1B). The LT_50_ of the 9:1 group was significantly lower (Tukey’s test, *p* = 0.0499) than that of the 1:0 group and significantly higher (Tukey’ test, *p* < 0.0001) than those of the 1:1, 1:9, and 0:1 groups. Moisture culture results demonstrated that *B. bassiana* proliferated on the corpses in the 1:1-, 1:9-, and 0:1-injected groups, while *M. rileyi* was predominant in the 1:0 group (Figure 1C). However, fungal growth on carcasses from the 9:1-injected group was indeterminate.

### 3.2. Sequencing Data Statistics

The minimum operational taxonomic units per sample were 17.33, with significant increases observed in the 9:1 and 1:1 groups (Figure 2A). The coverage of bacteria in all samples exceeded 99.99% (Appendix A). The rarefaction curves indicated that the sequencing depth was adequate, as evidenced by the curves plateauing, demonstrating sufficient coverage and depth (Appendix A).

### 3.3. Diversity and Composition of Gut Bacteria in S. litura

The Chao index of richness was higher in the 9:1 (Tukey’s test, *p* = 0.0035, *p* = 0.0156, and *p* = 0.0156) and 1:1 (Tukey’s test, *p* = 0.0014, *p* = 0.0057, and *p* = 0.0057) treatment groups compared to the 1:0, 1:9, and 0:1 groups (Figure 2B). The average variation degree, a microbial community stability index, decreased with the balance of the *M. rileyi* to *B. bassiana* (*v*/*v*) mixing ratio (Figure 2C). Principal component analysis was employed to assess significant differences in the microbial communities of the *S. litura* gut among the five groups. As depicted in Figure 2D, the three biological replicates of each sample clustered together, dividing the samples into five groups based on varying mixing proportions. At the phylum level, Firmicutes, Proteobacteria, and Actinobacteria were the dominant intestinal bacteria across all groups, with Firmicutes being the most prevalent phylum (Figure 2E). Moreover, all five treatment groups shared a common core microbiome, predominantly consisting of the Firmicutes phylum (Figure 2F,G). A small proportion of the operational taxonomic units found in the 0:1, 1:1, and 1:9 treatment groups corresponded to the Firmicutes, Proteobacteria, and Actinobacteria phyla, respectively. The primary groups of shared operational taxonomic units between the 9:1- and 1:1-injected groups were the Proteobacteria and Actinobacteria phyla. 

### 3.4. Abundance and Diversity Analyses of the Gut Bacteria in S. litura

Analysis of the top 10 species at the genus level revealed minor differences among the gut microbial communities of the five groups (Figure 3A). The dominant intestinal bacteria within these groups were *Enterococcus*, *Weissella*, *unclassified_f_Enterobacteriaceae*, and *Corynebacterium*, with *Enterococcus* being the predominant genus. Additionally, a heatmap displayed the top 20 genera (Figure 3B). The 1:0-injected group showed increased relative abundances of the *Achromobacter* and *Pseudomonas* genera. In the 9:1-injected group, the abundances of the *unclassified_o__Lactobacillales*, *Staphylococcus*, *Corynebacterium*, and *Pedobacter* genera increased, while that of the *Lactococcus* genus decreased. The 1:1-injected group exhibited increases in eight genera, i.e., *Lactococcus*, *Brucella*, *Glutamicibacter*, *Acinetobacter*, *Brevibacterium*, *Pediococcus*, *unclassified_f_Enterobacteriaceae*, and *Brachybacterium*, with a decrease in *Pedobacter*. The 1:9-injected group saw elevated levels of *Enterococcus*, *ZOR0006*, *Leuconostoc*, and *Turicibacter* and a reduction in *Lactobacillus*. In the 0:1-injected group, the *Weissella* genus increased, while *Enterococcus* and *Pseudomonas* decreased. To identify key microbes associated with different treatments, linear discriminant analysis (LDA) was performed on the microbial abundance profiles, identifying 3, 6, and 10 key microbes in the 1:0, 1:1, and 9:1 treatment groups, respectively, including *unclassified_f_Enterobacteriaceae*, *Corynebacterium*, *unclassified_o_Lactobacillales*, and *Staphylococcus* (Figure 3C).

### 3.5. Analysis of the Microbial Community and Composition of Gut Bacteria in S. litura

At the phylum level (Figure 4A), the relative abundance of the Firmicutes phylum was significantly higher (Tukey’s test, *p* < 0.0001 and *p* = 0.0005) in the 1:0 group compared to the 9:1 and 1:1 groups (Figure 4A). The Proteobacteria phylum showed an increase in the 9:1 (Tukey’s test, *p* = 0.4011 and *p* = 0.4022), 1:1 (Tukey’s test, *p* = 0.0045 and *p* = 0.0045), and 0:1 (Tukey’s test, *p* = 0.2815 and *p* = 0.9993) groups compared with the 1:0 and 1:9 groups. The abundance of the Actinobacteriota phylum was notably elevated in the 9:1 group (Tukey’s test, *p* = 0.0010, *p* = 0.0085, *p* = 0.0036, and *p* = 0.0006). At the genus level (Figure 4B), the 9:1-injected group displayed a significant increase in the relative abundances of *Staphylococcus*, *Corynebacterium*, and *unclassified_o_Lactobacillales* compared with the 1:0 and 0:1 groups. The relative abundances of *Brevibacterium*, *unclassified_f_Enterobacteriaceae*, *Brachybacterium*, *Glutamicibacter*, and *Brucella* were significantly higher in the 1:1 group than in the 1:0 and 0:1 groups. Moreover, the microbial community composition changed due to competition between *M. rileyi* and *B. bassiana* (Figure 4C and Appendix A). The complexity of the microbial community was lower in the 1:0 and 1:9 groups than in the 9:1, 1:1, and 0:1 groups.

### 3.6. Gut Microbial Community Functional Prediction for S. litura

To elucidate the functional capabilities of the gut microbial community in the oriental armyworm, we performed 16S rRNA sequencing using Tax4Fun. Functional prediction of the bacterial communities (Figure 5A) revealed significant differences in four pathways at Kyoto Encyclopedia of Genes and Genomes level 2 across the groups. The pathways of nucleotide metabolism, translation, and replication and repair were more abundant in the 1:0, 1:1, and 0:1 groups compared with the 9:1 and 1:9 groups. Conversely, cell motility pathways were more abundant in the 9:1 and 1:9 groups than in the 1:0, 1:1, and 0:1 groups. At Kyoto Encyclopedia of Genes and Genomes level 3 (Figure 5B), pathways such as pentose and glucuronate interconversions, purine metabolism, pyrimidine metabolism, lysine biosynthesis, glutathione metabolism, peptidoglycan biosynthesis, aminoacyl-tRNA biosynthesis, ribosome, nucleotide excision repair, mismatch repair, and homologous recombination were significantly more abundant in the 1:0, 1:1, and 0:1 groups compared with the 9:1 and 1:9 groups. Pathways like the phosphotransferase system (PTS), bacterial invasion of epithelial cells, ABC transporters, two-component system, bacterial chemotaxis, flagellar assembly, and porphyrin and chlorophyll metabolism were more abundant in the 9:1 and 1:9 groups than in the 1:0, 1:1, and 0:1 groups.

### 3.7. M. rileyi and B. bassiana in Multiple Stress Responses

*M. rileyi* demonstrated greater sensitivity to NaCl, H_2_O_2_, and menadione (MND) stressors during seven-day colony growth at 25 °C on Czapek–Dox medium plates supplemented with these chemical stressors compared to *B. bassiana* (Figure 6A,B). Additionally, the growth of hyphal bodies within the host hemocoel was observed (Figure 6C). By 60 h post-injection, *B. bassiana* began producing hyphal bodies from the host hemocoel, and by 132 h post-injection, *M. rileyi* also produced hyphal bodies, while *B. bassiana* generated numerous free-floating hyphal bodies. The results of the plate confrontation experiment of fungi showed that the colony morphology of *B. bassiana* was significantly inhibited by *M. rileyi* on the plate with a close distance from the starting growth point, while no similar phenomenon was shown in the control group (Figure 6D).

## 4. Discussion

*B. bassiana* and *M. rileyi* are known as entomophagous fungi that infect arthropods, including *S. litura*. The percentage mortality of *S. litura* third instar reached over 50% at 4 days for *B. bassiana*, with 1 × 10^8^ spores/mL, and the final mortality rate reached 100% [27]. The percentage mortality of *S. litura* second instar reached over 50% at 1.343 days for *M. rileyi*, with 1 × 10^8^ spores/mL, and the final mortality rate reached 100% [28]. In the present study, the LT_50_ of the 1:0 group was significantly higher than those of the 9:1, 1:1, 1:9, and 0:1 groups, showing that *B. bassiana* has high virulence for *S. litura.* Notably, *B. bassiana* also grew out from dead insect bodies in the 9:1-injected group, suggesting that *B. bassiana* has stronger competitiveness in the host’s hemolymph than *M. rileyi.*

Our findings indicate that there was a difference in the gut microbial communities after both *B. bassiana* and *M. rileyi* injection. *Enterococcus mundtii*, a symbiotic bacterium in the gut of *Spodoptera littoralis*, has been reported to secrete bactericin mundticin KS, a bacteriocin that selectively eliminates pathogenic bacteria within the host’s gut without affecting other members of the gut microbiome, thereby enhancing the stability of the intestinal microbial community [29]. *Wolbachia* endosymbionts in *Aedes mosquitoes* can induce oxidative stress, increase ROS levels, and activate the oxygen-dependent Toll pathway, which mediates the host’s antioxidant response. Concurrently, these endosymbionts produce antimicrobial peptides such as defensin and cecropin, which inhibit the replication of the dengue virus [30,31]. Notably, *Wolbachia* is widely distributed in insect tissues such as the midgut, Martensian ducts, and hemolymph [32], suggesting that the symbiotic bacterium can manipulate the host defense system to facilitate its own persistent infection. In the later stage of fungal infection, an increased abundance of *Serratia marcescens* in the insect gut can partially inhibit the virulence of *B. bassiana* [18]. *M. rileyi* infection induced the translocation of gut bacteria, and then the fungi activated and exploited its host humoral antibacterial immunity to eliminate opportunistic bacteria, preventing them from competing for nutrients in the hemolymph [20]. These observations underscore the importance of intestinal bacteria as vital biological agents that contribute to the balance of the intestinal bacterial community and play roles in host resistance or defense against the invasion of foreign species. 

AKP would be beneficial under changing environmental conditions as it would maximize chances that suitable bacteria, occurring within the microbial pool, could contribute to the insect responses to stress [33]. Studies have shown that the gut of Lepidoptera larvae harbors abundant microorganisms, particularly bacteria [34,35]. There are notable differences not only in the structure but also in the function of bacterial communities across different Lepidoptera species [36,37]. Comparative analyses by culture-based, cloning/sequencing, or high-throughput amplification have identified *Enterobacteriaceae*, *Bacillaceae*, and *Pseudomonadaceae* as the most widely distributed gut bacteria in these insects [38]. Our findings indicate significant increases in the abundances of *Achromobacter* and *Pseudomonas* in the intestines of larvae infected by *M. rileyi*, while infections by *B. bassiana* led to a significant increase in the relative abundance of the *Weissella* genus. Conversely, the abundances of *Enterococcus* and *Pseudomonas* decreased. Notably, the multivariate dispersion of microbial patterns is significantly higher under these stressful environmental conditions [33]. In this study, we found that the dispersion level was higher in the 1:0 and 0:1 groups than in the 9:1, 1:1, and 1:9 groups (Appendix A), implying that the pressure of individual infection by *M. rileyi* and *B. bassiana* was greater than that of the mixed group. These results suggest that larvae employ different strategies to cope with infections by *M. rileyi* and *B. bassiana*, indicating distinct pathogenic mechanisms for these fungi. This divergence may be attributed to the different pathogenic strategies the fungi employ during in vivo proliferation.

The ability of fungi to grow, develop, and resist stress may lead them to adopt various proliferation strategies in the host’s hemolymph. The process by which entomopathogenic fungi kill hosts involves complex biochemical interactions, including hydrophobic interactions, the interaction of body wall-degrading enzymes and toxins with the host, microbial degradation, and host responses such as behavioral fever, blood cell degradation, melanism, and immune responses [39]. When pathogenic bacteria invade, they inhibit host microbiota by secreting antibacterial substances to maintain a competitive advantage. For instance, hydroxyfungerins [40] secreted by *Metarhizium* sp. FKI-1079 and ergosterol peroxide [41] produced by *M. rileyi* exhibit cytotoxic and antibacterial activities, which also help eliminate other microorganisms colonizing the host. Conversely, *B. bassiana* rapidly colonizes the host’s hemolymph due to its quick growth and strong resistance [1,5]. Additionally, fungi exhibit varying stress resistance capabilities at different growth stages. Generally, *B. bassiana*’s growth ability surpasses that of *M. rileyi* [2,42]. While most cells in *M. rileyi*’s gut exist as spores, *M. rileyi* also presents mycelium that has undergone a dimorphic transformation, suggesting distinct survival strategies in the insect blood cavity compared to *B. bassiana*, which appears more resilient. Our findings align with this observation; *B. bassiana*’s spores were generally observed in the insect at 60 h, whereas *M. rileyi*’s spores appeared at 132 h, indicating a potentially more robust defense system in *B. bassiana*.

KEGG functional enrichment analysis revealed differences in the functional enrichment of intestinal microorganisms across different inoculation ratios. Pathways such as pentose and glucuronate interconversions, purine metabolism, pyrimidine metabolism, lysine biosynthesis, and homologous recombination were significantly more abundant in the 1:0, 1:1, and 0:1 groups compared to the 9:1 and 1:9 groups. Pathways associated with the phosphotransferase system (PTS), bacterial invasion of epithelial cells, ABC transporters, and porphyrin and chlorophyll metabolism were more prevalent in the 9:1 and 1:9 groups (Figure 5). These findings suggest that when two pathogenic fungi co-infect a host with varying inoculum sizes, the host’s intestinal bacteria employ similar strategies to resist infection, indicating a broadly similar defense system within the gut microbiome against fungal pathogens. When an equal proportion of the two fungi is used for infection, the host’s intestinal microbiome seems better equipped to defend against infection by a single pathogenic fungus (1:0 and 0:1 groups). These results underscore the different mechanisms by which *M. rileyi* and *B. bassiana* cause disease in insects.

Despite the positive outcomes, this study has limitations, including the need to further screen and validate the functions of inhibitory genes in *M. rileyi* and *B. bassiana* and to explore, as extensively as possible, the unique virulence genes specific to these two entomopathogenic fungi. It is anticipated that this will provide a more substantial scientific basis for the development and application of *M. rileyi* and *B. bassiana* biologics. In conclusion, our results preliminarily reveal the differences in pathogenicity between *M. rileyi* and *B. bassiana* from the perspective of gut microbial involvement in host defense. These findings enhance our understanding of the diverse pathogenic strategies of entomopathogenic fungi and could improve the biological control of agricultural and forestry pests.

## 5. Conclusions

Our findings indicate that the virulence of *B. bassiana* against larvae is significantly greater than that of *M. rileyi*, with survival analyses of the 1:9, 1:1, and 9:1 groups demonstrating that a higher proportion of *B. bassiana* in the fungal spore suspension correlates with a lower LT_50_ for the larvae. Analysis of gut microbial diversity revealed that infections with *M. rileyi*, *B. bassiana*, and three mixed groups (1:9, 1:1, and 9:1) led to significant increases in the abundances of bacteria across various genera. Early infection stages showed differences between *M. rileyi* and *B. bassiana* in the larvae of *S. litura*, potentially due to *B. bassiana*’s stronger growth and stress resistance, and the antifungal ability of *M. rileyi* was better than that of *B. bassiana*, as confirmed by our in vitro experiments. In summary, our study preliminarily confirms the distinctions in the mechanisms of different entomopathogenic fungi, thereby enhancing guidance for the biological control of agricultural and forestry pests.

## Figures and Tables

**Figure 1 microorganisms-12-01129-f001:**
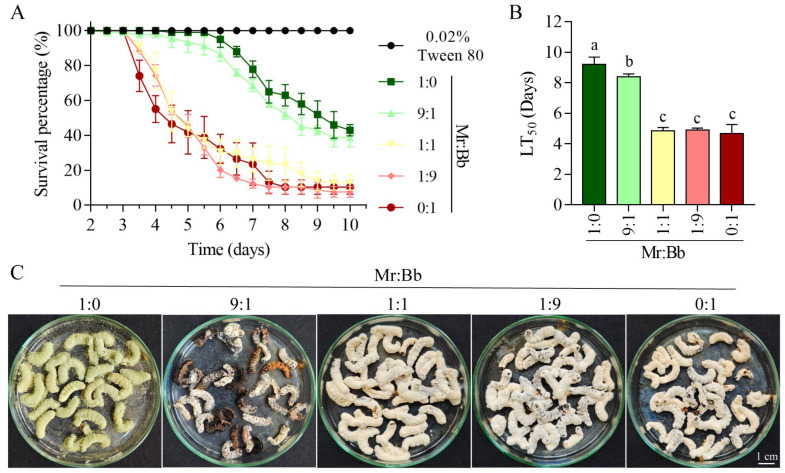
The virulence of *B. bassiana* AJS91881 and *M. rileyi* SXBN200920 to the larvae of *S. litura*. (**A**,**B**): Mortality and median lethal time (LT_50_) of larvae infected with spore suspensions of *B. bassiana* AJS91881 and *M. rileyi* SXBN200920 in five groups of 1 × 10^5^ spores/mL. Categories include 0.02% Tween 80, Mr:Bb (1:0), Mr:Bb (0:1), Mr: Bb (1:1), Mr: Bb (1:9), and Mr: Bb (9:1). (**C**): The state of moist culture for 10 days after the death of insects infected with fungal spore suspensions of the above 5 mixed groups. Different letters above the bars indicate statistical significance (*p* < 0.05). Error bar: standard deviation.

**Figure 2 microorganisms-12-01129-f002:**
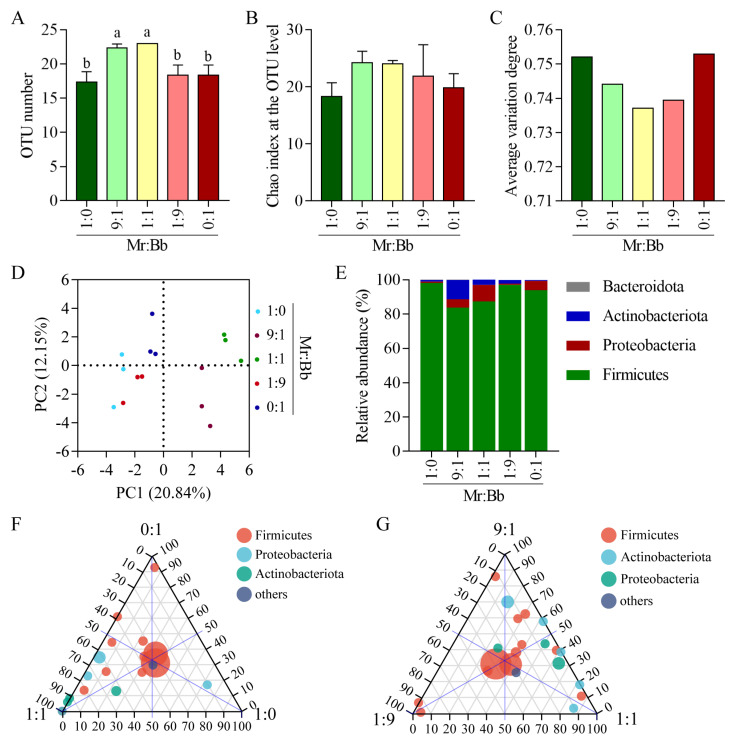
Effects of co-infection of *B. bassiana* and *M. rileyi* on the intestinal microbial diversity of *S. litura* larvae at the phylum level. (**A**–**C**): operational taxonomic unit index (**A**), Chao index (**B**) and average variation degree (**C**) at the phylum level. a and b represent statistically significant differences between groups. (**D**) Principal component analysis of the intestinal microbial β diversity index of *B. bassiana* AJS91881- and *M. rileyi* SXBN200920-infected insects with 1 × 10^5^ spores/mL spore suspension for 3 days. (**E**) The relative abundance of gut microbes in five groups of fungus-infected insects at the phylum level. (**F**) Ternary phase diagrams for groups 1:1, 0:1, and 1:0. (**G**) Ternary phase diagrams for groups 1:1, 9:1, and 1:9. The ternary phase diagram shows the changes in the composition and distribution of matter-dominant species in three different groups/samples and can visually analyze the abundance of the same species in different groups. The three corners represent three or three groups of samples, the solid circle in the figure represents the species below the level of the gate, and the size of the circle represents the average relative abundance of the species. Different letters above the bars indicate statistical significance (*p* < 0.05).

**Figure 3 microorganisms-12-01129-f003:**
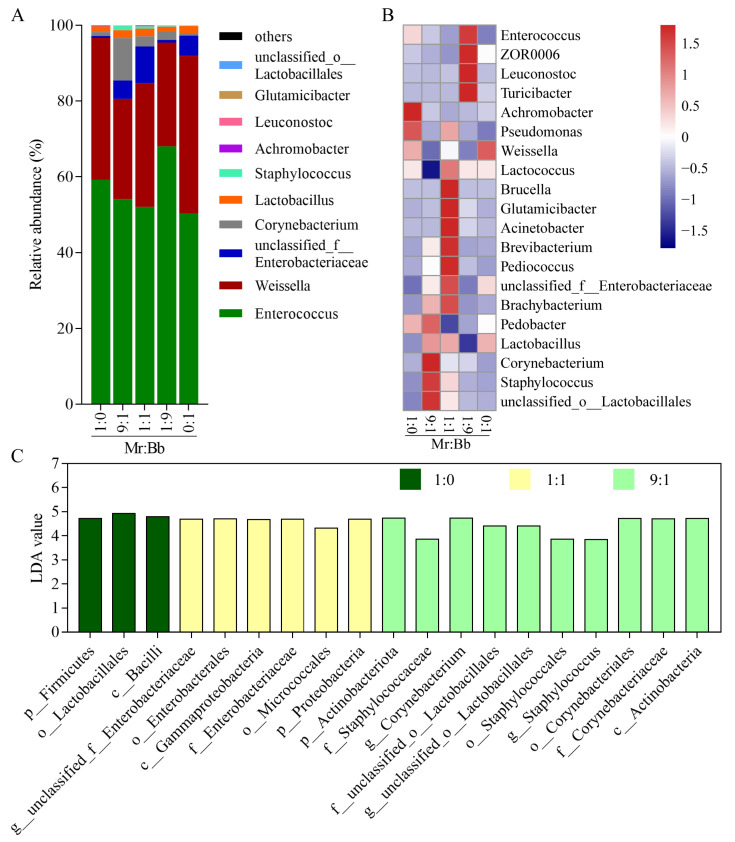
Effects of co-infection of *B. bassiana* and *M. rileyi* on the intestinal microbial diversity of *S. litura* larvae at the genus level. (**A**): Distribution of the top 10 microbial species in the gut of *S. litura* larvae infected by 5 groups of 1 × 10^5^ spores/mL of *B. bassiana* AJS91881 and *M. rileyi* SXBN200920 spore suspensions. (**B**): Heat maps of the relative abundance of the top 20 microorganisms in the gut of insects treated in the above 5 groups. (**C**): Analysis of insect gut microbial abundance profiles in three fungal treatment groups (1:0, 1:1, and 9:1).

**Figure 4 microorganisms-12-01129-f004:**
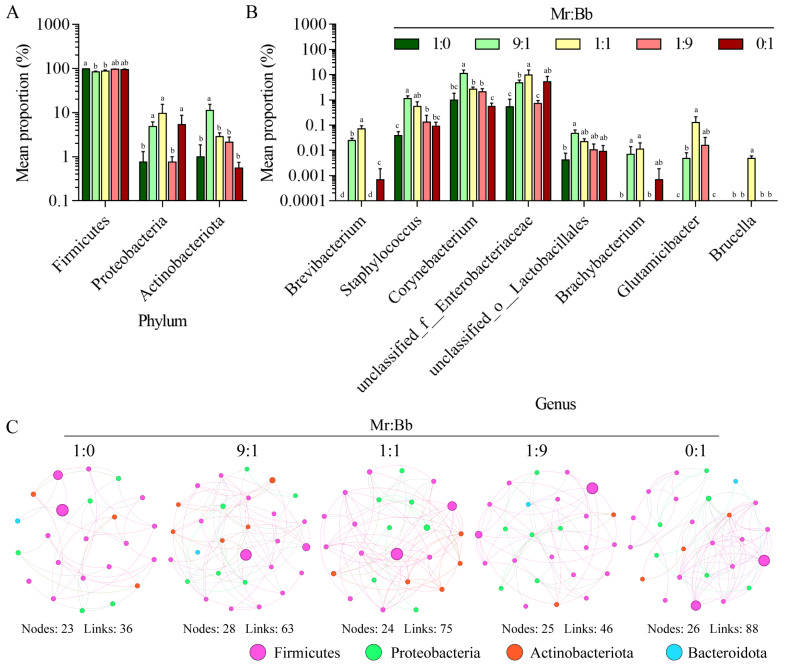
Changes and interaction analysis of intestinal flora of *S. litura* infected by *B. bassiana* and *M. rileyi*. (**A**,**B**) are the changes of intestinal flora at the phylum level and genus level, respectively. Different letters above the bars indicate statistical significance (*p* < 0.05). (**C**): In addition, the interaction network maps between the effects of the five fungal treatment groups on insect gut microbes were based on the gate level.

**Figure 5 microorganisms-12-01129-f005:**
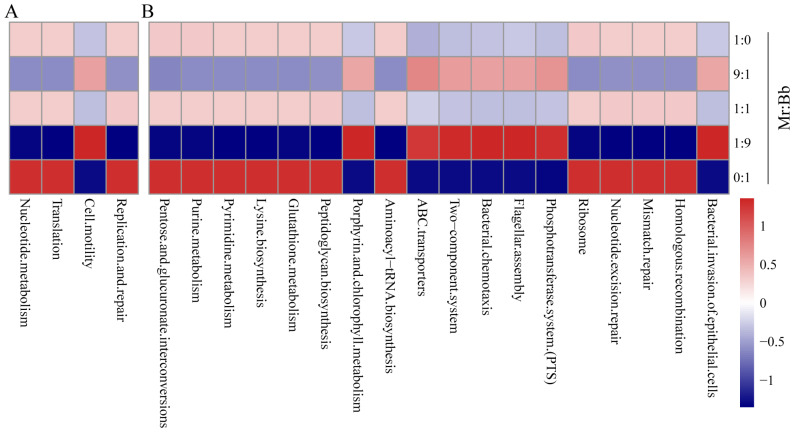
Prediction of intestinal microbial community function of *S. litura*. (**A**,**B**) are Kyoto Encyclopedia of Genes and Genomes function predictions at levels 2 and 3, respectively. The Kyoto Encyclopedia of Genes and Genomes pathways analyzed include various metabolic pathways, synthetic pathways, membrane transport, signaling, cell cycle, and disease-related pathways.

**Figure 6 microorganisms-12-01129-f006:**
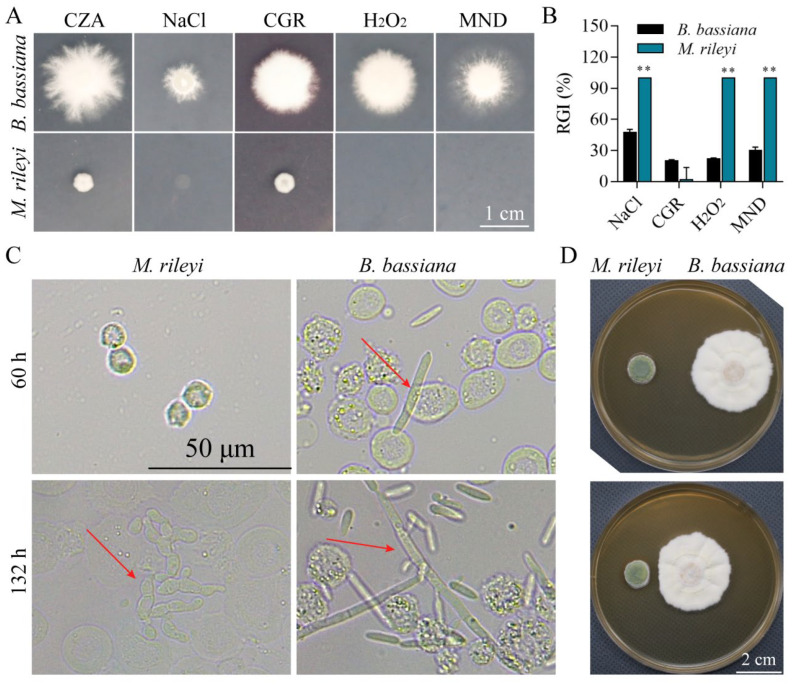
Comparison of stress resistance between *B. bassiana* and *M. rileyi*. (**A**): Growth status of 1 × 10^6^ spores/mL of *B. bassiana* AJS91881 and *M. rileyi* SXBN200920 on different chemical stress media, on day 7. Among them, stressors include sodium chloride (NaCl), Congo red (CGR), hydrogen peroxide (H_2_O_2_), and menadione (MND). (**B**): Colony diameters of two strains of fungi on different chemical stress media. Tukey’s HSD test revealed significant differences: *p* < 0.05. Error bar: standard deviation. *p* < 0.01 (**). (**C**): Fungal spore growth in the insect blood cavity of *S. litura* third instar larvae at 60 h and 132 h after injection of 1 × 10^5^ spores/mL of *B. bassiana* AJS91881 and *M. rileyi* SXBN200920, respectively. The red arrow represents the fungal spore in the blood cavity. (**D**): Growth confrontation between *B. bassiana* AJS91881 and *M. rileyi* SXBN200920 on SDAY plate. The starting points of colony growth were 4.5 cm (**top**) and 2 cm (**bottom**).

## Data Availability

The original contributions presented in the study are included in the article/Appendix A, further inquiries can be directed to the corresponding authors.

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
