# Peer review of "Gut Microbial Diversity Reveals Differences in Pathogenicity between Metarhizium rileyi and Beauveria bassiana during the Early Stage of Infection in Spodoptera litura Larvae"

_microorganisms, 2024, doi:10.3390/microorganisms12061129_

Round 1

Reviewer 1 Report

Comments and Suggestions for Authors

Manuscript (MS, hereafter) reports important data on the pathogenicity of M. rileyi and B. bassiana to larvae of S. litura and their effect on hots gut microbial composition and prevalence. In brief, results showed that larvae of S. litura expressed significant differences in its sensitivity to M. rileyi and B. bassiana and there is significant difference in gut symbionts composition between larvae infected by different ratio of M. rileyi and B. bassiana. The topic is worthy of investigation. However, I feel that MS suffers from relevant weaknesses, particularly regarding the materials and methods, the result, and discussion sections. The overall quality of the MS should be strongly improved before it can be considered for publication on microorganisms.

Some of the specific comments that I have are indicated in the MS

Introduction is narrow for an international journal. They should be expanded according to the topic and the key words of the paper. I suggest to added a short part to refer to possible links with Anna Karenina principal.

Material and methods are poorly written and lacking in technical details in several sections. In addition, the absence of negative control with non-infected larvae make interpretation of results very difficult and inconclusive. The statistical analysis is not clear and should be re-written including more details: the analysis used for each parameter and if there are some transformations...?

Results are descriptive and even in the sections where a statistical test has been carried out, the P values and test are not given.

Discussion is too scanty. The main criticism I noted is that Authors have touched several issues without developing properly the consequences of their findings. I was tempted to suggest adding a part on multivariate dispersion (see paper on «Anna karenina» principle).

Comments on the Quality of English Language

N.A.

Author Response

Reply to Reviewer1

Dear Reviewers:

Thank you for giving us an opportunity to revise our manuscript and your positive and constructive comments and suggestions on our manuscript. We have studied reviewer’s comments carefully and tried our best to revise our manuscript according to the comments. We respond to the reviewers comments point by point below. The changes are also marked in the revised manuscript with track.

Introduction is narrow for an international journal. They should be expanded according to the topic and the key words of the paper. I suggest to added a short part to refer to possible links with Anna Karenina principal.

Response: Thanks for your critical review and valuable suggestion. We have added this information in the Introduction part (Line 71-74). For more details, please refer to the revised manuscript and PDF file “microorganisms-3013210-res-2.0.PDF”.

Material and methods are poorly written and lacking in technical details in several sections. In addition, the absence of negative control with non-infected larvae make interpretation of results very difficult and inconclusive. The statistical analysis is not clear and should be re-written including more details: the analysis used for each parameter and if there are some transformations...?

Response: Thanks for your critical review; we have provided the details on the Methods and Data analysis parts. For more details, please refer to the revised manuscript and PDF file “microorganisms-3013210-res-2.0.PDF”.

Results are descriptive and even in the sections where a statistical test has been carried out, the P values and test are not given.

Response: We fixed this in our revised manuscript. For more details, please refer to the revised manuscript and PDF file “microorganisms-3013210-res-2.0.PDF”.

Discussion is too scanty. The main criticism I noted is that Authors have touched several issues without developing properly the consequences of their findings. I was tempted to suggest adding a part on multivariate dispersion (see paper on «Anna karenina» principle).

Response: Thanks for your valuable suggestion. We have added the information in the manuscript (Line 369-378 and line 411-418) and provided the PCoA analysis in the new Figure S2. For more details, please refer to the revised manuscript and PDF file “microorganisms-3013210-res-2.0.PDF”.

Reviewer 2 Report

Comments and Suggestions for Authors

The work is simple, but it is enlightening and brings new knowledge between two species of entomopathogenic fungi.

For culturing, Sabouraud Maltose Agar medium plates (SMAY: 1% peptone, 1% yeast extract, 4% maltose, and 1.5% agarose) and Sabouraud Dextrose Agar medium plates (SDAY: 1% peptone, 1% yeast extract, 4% glucose, and 1.5% agar) were used.

These media are protein-rich. Why were they chosen, and are there better ways? Is there any similar work? What other media have been used in similar work?

In the representation of results, Figure titles should be self-explanatory and without abbreviations. The reader has to interpret and understand everything. Levels? Species?.…

In line 358- Comparative analyses have identified Enterobacteriaceae, Bacillaceae, and Pseudo- monadaceae as the most widely distributed gut bacteria in these insects. Was the methodology the same?

In line 347- Wolbachia endosymbionts in Aedes mosquitoes can induce oxidative stress, increase ROS levels, and activate the oxygen-dependent Toll pathway, which mediates the host's antioxidant response. In this context, why did you put this bacterium?

Overall, the work is well completed.

Author Response

Reply to Rviewer 2

Dear Reviewers:

Thank you for giving us an opportunity to revise our manuscript and your positive and constructive comments and suggestions on our manuscript. We have studied reviewer’s comments carefully and tried our best to revise our manuscript according to the comments. We respond to the reviewers comments point by point below. The changes are also marked in the revised manuscript with track.

For culturing, Sabouraud Maltose Agar medium plates (SMAY: 1% peptone, 1% yeast extract, 4% maltose, and 1.5% agarose) and Sabouraud Dextrose Agar medium plates (SDAY: 1% peptone, 1% yeast extract, 4% glucose, and 1.5% agar) were used.

These media are protein-rich. Why were they chosen, and are there better ways? Is there any similar work? What other media have been used in similar work?

Response: Thanks for your critical review and valuable suggestion. M. rileyi can be found in SMAY, 1/4 SDAY, and PDA mediums cultivation, and B. bassiana can be found SDAY, 1/4 SDAY, and PDA mediums cultivation. M. rileyi and B. bassiana is usually cultured in SMAY (Lin et al., 2021) and SDAY (Celar and Kos, 2016), respectively. In order to reduce the production cost of conidia, other substance culture media have also been developed. Such as, M. rileyi was cultured in RVOSM (100 g rice and 3 mL vegetable oil) (Xue et al., 2023), and B. bassiana was cultured in white rice with 100% water yielded the highest conidial production (approximately 1.3 × 1010 conidia/g of substrate) (Rangel et al., 2023), suggesting that using culture media with rice substrates can produce spores on a large scale. We added these references in the revised manuscript.

In the representation of results, Figure titles should be self-explanatory and without abbreviations. The reader has to interpret and understand everything. Levels? Species?.…

Response: We fixed this in our revised manuscript.

In line 358- Comparative analyses have identified Enterobacteriaceae, Bacillaceae, and Pseudo- monadaceae as the most widely distributed gut bacteria in these insects. Was the methodology the same?

Response: Thank the reviewer’s very careful review. Gut microbiota in Lepidoptera was collected by using multiple methodology, including culture-based, cloning/sequencing, or high-throughput amplicon. This information has been added in the discussion part (Line 388-395).

In line 347- Wolbachia endosymbionts in Aedes mosquitoes can induce oxidative stress, increase ROS levels, and activate the oxygen-dependent Toll pathway, which mediates the host's antioxidant response. In this context, why did you put this bacterium?

Response: Thanks for your critical review and valuable suggestion. XL Pan et al. reported that Wolbachia endosymbionts in Aedes mosquitoes can induce reactive oxygen species (ROS)-dependent activation of the Toll pathway (Pan et al., 2012). Wolbachia is widely distributed in insect tissues such as the midgut, Martensian ducts, and hemolymph (Werren and Windsor, 2000). In entomophagous fungus, M. rileyi infection induced the translocation of gut bacteria, and then the fungi activated and exploited its host humoral antibacterial immunity to eliminate opportunistic bacteria, preventing them from competing for nutrients in the hemolymph (Wang et al., 2023). Thus, gut microbiome can manipulate the host defense system to facilitate its own persistent infection (Pan et al., 2012).

Overall, the work is well completed.

Response: Thank you very much for your affirmation of our work.

References

CELAR F A, KOS K 2016. Effects of selected herbicides and fungicides on growth, sporulation and conidial germination of entomopathogenic fungus Beauveria bassiana. Pest Manag Sci [J], 72: 2110-2117.

LIN Y, WANG J, YANG K, et al. 2021. Regulation of conidiation, polarity growth, and pathogenicity by MrSte12 transcription factor in entomopathogenic fungus, Metarhizium rileyi. Fungal Genet Biol [J], 155: 103612.

PAN X, ZHOU G, WU J, et al. 2012. Wolbachia induces reactive oxygen species (ROS)-dependent activation of the Toll pathway to control dengue virus in the mosquito Aedes aegypti. Proc Natl Acad Sci U S A [J], 109: E23-31.

RANGEL D E N, ACHEAMPONG M A, BIGNAYAN H G, et al. 2023. Conidial mass production of entomopathogenic fungi and tolerance of their mass-produced conidia to UV-B radiation and heat. Fungal Biol [J], 127: 1524-1533.

WANG J L, SUN J, SONG Y J, et al. 2023. An entomopathogenic fungus exploits its host humoral antibacterial immunity to minimize bacterial competition in the hemolymph. Microbiome [J], 11: 116.

WERREN J H, WINDSOR D M 2000. Wolbachia infection frequencies in insects: evidence of a global equilibrium? Proc Biol Sci [J], 267: 1277-1285.

XUE R, DU G, CHEN C, et al. 2023. Oleic acid improves the conidial production and quality of Metarhizium rileyi as a biocontrol agent. Biocontrol Science and Technology [J], 33: 758-771.

Round 2

Reviewer 1 Report

Comments and Suggestions for Authors

All comments were considered by authors and the quality of manuscript was improved.